# DEEP HIERARCHICAL-HYPERSPHERICAL LEARNING (DH²L)

## ABSTRACT

Regularization is known to be an inexpensive and reasonable solution to alleviate the overfitting problem arising in inference models, including deep neural networks. In this paper, we propose hierarchical regularization which preserves the semantic structure of a sample distribution. At the same time, this regularization promotes diversity by imposing a distance between parameter vectors enlarged within semantic structures. To generate evenly distributed parameters which is considered less redundant, we constrain them to lie on *hierarchical hyperspheres*. To define a hierarchical parameter space, we propose to reformulate the topology space with multiple hyperspheres. On each hypersphere, a projection is parameterized by two individual parameters. Since maximizing a groupwise and pairwise distance between points on hypersphere is nontrivial (generalized Thomson problem), we propose a new discrete metric integrated with a continuous metric. The proposed method shows improved generalization performance on extensive experiments using publicly available datasets (CIFAR-10, CIFAR-100, CUB200-2011, and Stanford Cars), especially when the number of super-classes is large.

## 1 INTRODUCTION

Diversity promoting learning has been widely adopted via enlarging pairwise distances (Xie et al., 2018; 2017a; Liu et al., 2018), increasing orthogonality (Xie et al., 2018), reducing covariance between parameters (Xie et al., 2017b), or reducing correlation on feature (Cogswell et al., 2016) to improve generalization performance. Among them, *diversity promoting regularization* (Xie et al., 2017a;b; Liu et al., 2018) by enforcing large diversity between projection parameters achieves a reasonable performance without modifying the model structure. While optimizing the objective function with a covariance matrix in (Xie et al., 2017b; 2018) is nontrivial, the diversity promoting regularization via minimizing energy of parameters of deep neural networks has been proposed (Liu et al., 2018) in a simpler way. By minimizing a pairwise distance between parameters on a hypersphere with the known metric, they achieved the improved generalization performance.

Following an efficient regularization on the hypersphere, we explore further this direction with three main concepts (hierarchical and hyperspherical learning with discrete metrics).
1) *Why hierarchical learning?* Hierarchical inference explains *human intelligence*. In (Kurzweil, 2013), it states that "the neocortex contains about 300 million very general pattern recognizers, arranged in a hierarchy". Applying the hierarchy of multiple classes based on semantic taxonomy is a natural choice to devise *machine intelligence*. Effectiveness of the hierarchical learning can be found in (Verma et al., 2012).

2) *Why hyperspherical learning?* Hypersphere is the set of points at the equidistance (radius) from a given point (centroid) in a certain dimensional space. Due to the denominator in the unit-length normalization ($\frac{w}{\|w\|}$, $w \in \mathbb{R}^{d+1}$), the angular distance defined on the hypersphere converges when the magnitude of $w$ goes infinity while Euclidean distance goes infinity. Due to this bounded property, a hierarchical structure with multiple separated hyperspheres can be defined.

3) *Why discrete metric learning?* If vector points form discontinuous series with discrete representation (e.g. multi-dimensional binary or ternary), they are isolated from each other with a certain margin. This property may fit with a disconnected/groupwise manifold space problem. We note that making points to be equidistributed where a pairwise distance is maximized is a nontrivial task. As

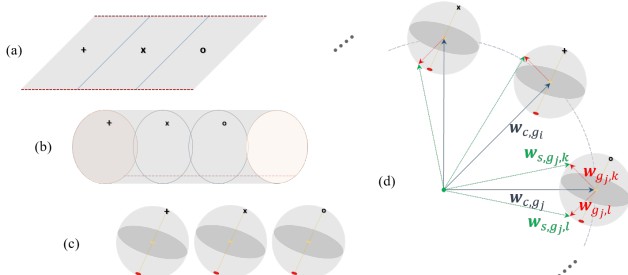

Figure 1: Multiple (hyper)spheres as quotient spaces of a topology space on Euclidean space might be found by gluing process with identifying points. Those separated hyperspheres are assumed to be under the quotient space conditions (Tu, 2010). Within an individual (hyper)sphere, projection parameters in deep neural networks preserve a hierarchical structure. The space can be formed in a series: (a), (b), (c), and (d)

points in a discrete metric space are finite and isolated, search efforts to satisfy those constraints can be reduced.

In this paper, we propose to apply a hierarchical structure to parameter regularization on the multiple groupwise hyperspheres. In order to find an appropriate metric in this space, we explore a discrete angular metric. We examine the proposed method on extensive experimental setups in terms of datasets and deep network models.

## 2  MULTIPLE SEPARATED HYPERSPHERES

Samples observed from the real world may be on disconnected manifolds. In other words, the disjoint union of those manifolds can generate the global manifold (Lee, 2000). In this section, we decompose the one space into multiple spaces (manifolds) and re-define the spaces in a hierarchical point of view.

### 2.1  DISCONNECTED MANIFOLD VIA EQUIVALENT RELATIONS

Since it is not suitable to measure a pairwise distance between high dimensional vectors which have the hierarchical structure in the same space, we construct another identification space where subspaces (sub-manifolds) are isolated (via equivalence relation (Tu, 2010)). Denote $d$-sphere $\mathbb{S}^d$ to be the set of points that satisfies $\mathbb{S}^d = \{\boldsymbol{w} \in \mathbb{R}^{d+1} : \|\boldsymbol{w}\| = 1\}$. We construct multiple separated hyperspheres using multiple identifying relations. In Figure 1, we use a center parameter and a surface parameter to define a projection parameter vector $\boldsymbol{w}$ which constitutes a hypershpere space.

### 2.2  PRIOR DISTRIBUTION AND REGULARIZATION

To make parameter vectors uniformly distributed on the unit hypersphere, parameters are sampled from the Gaussian normal distribution (Muller, 1959; Harman & Lacko, 2010). This is because the normal distribution is spherically symmetric (Muller, 1959). In a Bayesian point of view, neural networks with Gaussian priors are known to induce $l^2$-norm regularization (Vladimirova et al., 2019). From two evidences, we know that enforcing parameters to have the Gaussian prior is important for hyperspherical learning in neural networks. Note that a parameter which is calculated from the difference arithmetic operation with two parameters on the normal Gaussian distribution is on the *normal difference distribution*.

## 3  METHOD

In deep neural networks, the objective function $\mathcal{J}$ with regularization $\mathcal{R}$ in addition to a loss $\mathcal{L}$, $\mathcal{J}_{\mathcal{R}(\mathbf{W})} = \mathcal{L}(\boldsymbol{x}, \mathbf{W}) + \lambda \mathcal{R}(\mathbf{W})$, is optimized to find the optimal $\mathbf{W}$ having a near minimum loss $\mathcal{L}$, $\arg \min_{\mathbf{W}} \mathcal{J}_{\mathcal{R}(\boldsymbol{x}, \mathbf{W})}$, where $\boldsymbol{x} \in \mathbb{R}^{d_0}$ denotes an input vector, $\mathbf{W} = \{\boldsymbol{W}_i \in \mathbb{R}^{d_{i-1} \times c_i} : \boldsymbol{W}_i = \{\boldsymbol{w}_j \in \mathbb{R}^{d_{i-1}}\}, j = 1, \dots, c_i, i = 1, \dots, L\}$ denotes a set of parameter matrices (i.e. neurons/kernels), $L$ denotes the number of layers, and $\lambda > 0$ is to control the degree of the regularization. For a classification task, the cross entropy loss is used for the loss function $\mathcal{L}$. We propose a new regularization formulation $\mathcal{R}$ preserving a hierarchical parameter structure in Section 3.1.

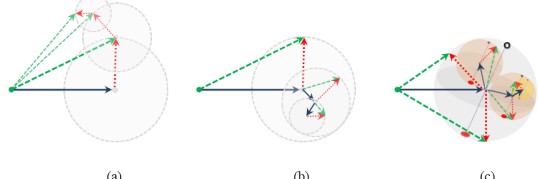

(a)            (b)            (c)

Figure 2: A series of levelwise ($l$) spheres with a radius ($r_l$) is shown: (a) A radius of an oveall area converges to $\frac{r_0}{1-\delta}$ ($= \sum_l^\infty r_0 \delta^l$: the sum of radius series) as $l$ goes to infinity where $l$ denotes a level, $r_0$ is their initial radius, and $\delta$ is the ratio between radiuses ($r_l, r_{l-1}$), $\frac{r_l}{r_{l-1}} < 1$. (b) The radius of the overall area is bounded to the initial radius $r_0$ of spheres. This bears a resemblance to the process of repeat of *Hypersphere packing* which arranges non-overlapping spheres within a containing space. (c) 2-sphere is defined following (b) which is appropriate to model a hierarchical structure. This can be generalized with a hypersphere ($\mathbb{S}^d$, $d \geq 3$) in a higher dimensional space.

## 3.1 REGULARIZATION FOR HIERARCHICAL AND HYPERSPHERICAL HYPOTHESES

Denote $\boldsymbol{w}$ a parameter vector (called a projection parameter, an element of $\boldsymbol{W}$ at a certain layer) to transform a given input into an embedding space defined in a Euclidean metric space: $\boldsymbol{x} \in \mathbb{R}^{d+1} \mapsto \boldsymbol{w}^T \boldsymbol{x} \in \mathbb{R}$. By the definition of unit-length projection $\frac{\boldsymbol{w}}{\|\boldsymbol{w}\|}$, a new parameter vector $\hat{\boldsymbol{w}}$ can be defined on $d$-sphere: $\mathbb{S}^d = \{\hat{\boldsymbol{w}} \in \mathbb{R}^{d+1} : \|\hat{\boldsymbol{w}}\| = 1\}$ where $\|\cdot\|$ denotes $l^2$-norm and a center is zero. In other word, the parameter $\hat{\boldsymbol{w}}$ can be defined by a center parameter vector $\boldsymbol{w}_c \in \mathbb{R}^{d+1}$ and a surface parameter vector $\boldsymbol{w}_s \in \mathbb{R}^{d+1}$ using an arithmetic operation: $\hat{\boldsymbol{w}} := \boldsymbol{w}_s - \boldsymbol{w}_c$. We define the $d$-sphere with the center and surface parameters: $\mathbb{S}^d_{\boldsymbol{w}_c} = \{\boldsymbol{w}_s - \boldsymbol{w}_c \in \mathbb{R}^{d+1} : \|\boldsymbol{w}_s - \boldsymbol{w}_c\| = 1\}$. For a notation simplicity, we use $\boldsymbol{w}$ instead of $\hat{\boldsymbol{w}}$ hereafter. While we consider a radius equals to 1 for simplicity, the parameter vector can have a radius $r > 0$.

### 3.1.1 HIERARCHICAL PARAMETERIZATION VIA LEVELWISE AND GROUPWISE PARAMETERS

We assume that the hierarchical structure consists of a levelwise structure with a notation ($l$) and a groupwise structure with a notation $g$ below. Through defining these structures serially, we explain hierarchical parameters.

**Levelwise structure**    The projection parameter $\boldsymbol{w} \in \mathbb{R}^{d+1}$ on $\mathbb{S}^d_{\boldsymbol{w}_c}$ can be defined with a levelwise notation ($l$) as follows,

$$\boldsymbol{w}^{(l)} := \boldsymbol{w}_s^{(l)} - \boldsymbol{w}_c^{(l)} \tag{1}$$

where $\mathbb{S}^d_{\boldsymbol{w}_c^{(l)}}$ denotes the $l$-th level $d$-sphere centered at $\boldsymbol{w}_c^{(l)}$. In this paper, we define the parameter in a higher dimensional space than that shown in Figure 2.

In a levelwise setting, $\boldsymbol{w}_s^{(l)}$ and $\boldsymbol{w}_c^{(l)}$ at the current level ($l$) are additively represented based on a center parameter of the previous level ($l-1$): $\boldsymbol{w}_c^{(l-1)} + \overrightarrow{\Delta \boldsymbol{w}}^{(l)} \mapsto \boldsymbol{w}_c^{(l)}$, where $\boldsymbol{w}_c^{(l-1)} = \sum_i^{l-1} \overrightarrow{\Delta \boldsymbol{w}}^{(i)}$ is the accumulated and $\overrightarrow{\Delta \boldsymbol{w}}^{(l)}$ denotes a connecting parameter from $\boldsymbol{w}_c^{(l-1)}$ to $\boldsymbol{w}_c^{(l)}$. By denoting $\overrightarrow{\Delta \boldsymbol{w}}^{(l)}$ as $\boldsymbol{w}^{(l,l-1)}$, the center vector at the $l$-th level is defined as, $\boldsymbol{w}_c^{(l)} := \boldsymbol{w}_c^{(l,l-1)} + \boldsymbol{w}_c^{(l-1)}$, and the surface vector as $\boldsymbol{w}_s^{(l)} := \boldsymbol{w}_s^{(l,l-1)} + \boldsymbol{w}_c^{(l-1)}$. Both the center parameter and the surface parameter at the current level are based on the center parameter at the previous level[1]. Hence, Eq. 1 is equivalent to

$$\boldsymbol{w}^{(l)} = \boldsymbol{w}_s^{(l,l-1)} - \boldsymbol{w}_c^{(l,l-1)}. \tag{2}$$

Note that we use a superscript ($l, l-1$) to denote parameters at the $l$-th level connected from the center parameter at the ($l-1$)-th level.

**Groupwise structure**    With a group notation $g_k$, the center parameter in Eq. 2 can be rephrased as $\boldsymbol{w}_{c,g_k}^{(l,l-1)}$, on $\mathbb{S}^d_{\boldsymbol{w}_{c,g_k}^{(l,l-1)}}$ which denotes $d$-sphere of $g_k$ group at the $l$-th level. Each group constitutes a group set at the $l$-th level $g^{(l)} := \{g_k\}_{k=1}^{|g^{(l)}|}$ where $g^{(l)} \subseteq \mathbb{G}^{(l)}$, $\mathbb{G}^{(l)}$ denotes a batch group set at the $l$-th level, and $|\cdot|$ denotes the cardinality. The group set $g^{(l)}$ at the $l$-th level is conditioned on

---

[1]As not every sample has a child sample, it might be more reasonable to branch from representative parameter or center parameter rather than from individual projection parameters.

a group set at the $(l-1)$-th level, $g^{(l-1)} := \{g_{k'}\}_{k'=1}^{|g^{(l-1)}|}$ where $g^{(l-1)} \subseteq \mathbb{G}^{(l-1)}$. By their group-wise relation over levels, an adjacency indication[2] $P^{(l,l-1)}(\{\mathbb{G}^{(l-1)}, \mathbb{G}^{(l)}\}) \in \{0,1\}^{|\mathbb{G}^{(l-1)}| \times |\mathbb{G}^{(l)}|}$ is given. Hence, the $i$-th parameter vector on $\mathbb{S}^d_{\boldsymbol{w}_{c,g_k}^{(l,l-1)}}$ is defined as follows,

$$\boldsymbol{w}_{g_k,i}^{(l)} := \boldsymbol{w}_{s,g_k,i}^{(l,l-1)} - \boldsymbol{w}_{c,g_k}^{(l,l-1)}, \tag{3}$$

where $\{\boldsymbol{w}_{s,g_k,i}^{(l,l-1)}, \boldsymbol{w}_{c,g_k}^{(l,l-1)}\}\ \forall i$ is calculated based on a center vector $\boldsymbol{w}_{c,g_{k'}}^{(l-1)}$ at the $(l-1)$-th level[3], $i = 1, \ldots, |g_k|$, and $|g_k|$ denotes the number of surface parameter vectors in the group $g_k$.

### 3.1.2 Hierarchical regularization

In this section, we define a regularization term using the hierarchical parameters defined above. A set of parameters $\{\boldsymbol{W}_{s,g_k}^{(l,l-1)}, \boldsymbol{w}_{c,g_k}^{(l,l-1)}, \boldsymbol{w}_{c,g_k'}^{(l-1)}\} \in \boldsymbol{W}\ \forall g_k,\ \forall g_{k'}$ where $\boldsymbol{W}_{s,g_k}^{(l,l-1)} := \{\boldsymbol{w}_{s,g_k,i}^{(l,l-1)}\}_{i=1}^{|g_k|}$, is an optimizing target of hierarchical regularization as follows:

$$\mathcal{R}(\boldsymbol{W}) := \sum_l \lambda_l \mathcal{R}_l(\boldsymbol{W}_{s,g_k}^{(l,l-1)}, \boldsymbol{w}_{c,g_k}^{(l,l-1)}; P^{(l,l-1)}) + \sum_l \mathcal{C}_l(\boldsymbol{w}_{c,g_k}^{(l,l-1)}, \boldsymbol{w}_{c,g_k'}^{(l-1)}; P^{(l,l-1)}) \tag{4}$$

where $\mathcal{R}_l$ works on individual spheres $\mathbb{S}^d_{\boldsymbol{w}_{c,g_k}^{(l,l-1)}}$, $\lambda_l > 0$, and $\mathcal{C}_l$ aims to apply geometry-aware constraints across spheres. $\mathcal{R}_l$ consists of two terms for regularization: 1) $\mathcal{R}_{l,p}$ for projection parameters in the same group $g_k$ on $\mathbb{S}^d_{\boldsymbol{w}_{c,g_k}^{(l,l-1)}}$ and 2) $\mathcal{R}_{l,c}$ for center parameters across groups in the same level on $\mathbb{S}^d_{\boldsymbol{w}_{c,g_k'}^{(l-1)}}$,

$$\mathcal{R}_l(\boldsymbol{W}_{s,g_k}^{(l,l-1)}, \boldsymbol{w}_{c,g_k}^{(l,l-1)}) := \mathcal{R}_{l,p}(\boldsymbol{W}_{s,g_k}^{(l,l-1)}, \boldsymbol{w}_{c,g_k}^{(l,l-1)}) + \mathcal{R}_{l,c}(\boldsymbol{w}_{c,g_k}^{(l,l-1)}), \tag{5}$$

where

$$\mathcal{R}_{l,p}(\boldsymbol{W}_{s,g_k}^{(l,l-1)}, \boldsymbol{w}_{c,g_k}^{(l,l-1)}) := \frac{1}{G} \sum_{\{g_k \in g^{(l)}\}} \frac{2}{G_k(G_k - 1)} \sum_{\{i \neq j \in g_k\}} d(\boldsymbol{w}_{g_k,i}^{(l,l-1)}, \boldsymbol{w}_{g_k,j}^{(l,l-1)}), \tag{6}$$

$$\mathcal{R}_{l,c}(\boldsymbol{w}_{c,g_k}^{(l,l-1)}) := \frac{2}{C(C-1)} \sum_{\{g_i \neq g_j \in g^{(l)}\}} d(\boldsymbol{w}_{c,g_i}^{(l,l-1)}, \boldsymbol{w}_{c,g_j}^{(l,l-1)}), \tag{7}$$

where $\boldsymbol{w}_{g_k,i}^{(l,l-1)} := \boldsymbol{w}_{s,g_k,i}^{(l,l-1)} - \boldsymbol{w}_{c,g_k}^{(l,l-1)}$, ($\boldsymbol{w}_{g_k,j}^{(l,l-1)}$ is similarly defined), $G = |\{g_k \in g^{(l)}\}|$, $G_k = |\{i \neq j \in g_k\}|$, $C = |\{g_i \neq g_j \in g^{(l)}\}|$, and $d(\cdot, \cdot)$ denotes a distance metric between parameters. The distance metric is defined in Section 3.2. We explain regularization terms formulated using these parameters and pairwise distances further in Figure 4 of Appendix. When a minibatch of inputs ($m_{\boldsymbol{x}}$: a set of inputs $\{\boldsymbol{x}_i\}$) is given, the regularization term becomes: $\mathbb{E}(\mathcal{R}(\boldsymbol{W})) = \frac{1}{|m_{\boldsymbol{x}}|} \sum_{m_{\boldsymbol{x}}} \mathcal{R}(\boldsymbol{W}; m_{\boldsymbol{x}})$.

The constraint term help construct geometry-aware relational parameters between different spheres on the same level and on the across levels. Multiple constraints are defined as $\mathcal{C}_l := \sum_k \lambda_k \mathcal{C}_{l,k}$, where $\mathcal{C}_{l,k}$ is the $k$-th constraint between parameters at the $l$-th and the $(l-1)$-th level, and $\lambda_k > 0$ is a Lagrange multiplier. We propose three constraints (defined in Appendix) in a geometric point of view.

### 3.2 Discrete and continuous angular distance metric

Discrete metric is a good fit for the above groupwise definition. We expect that points projected by parameters in a discrete metric space are isolated each other. In Figure 3, we show that discrete distances of parameter pairs have different values while their continuous distances are same. Hence, maximization of discrete distances of parameter pairs could help parameters distributed isolatedly and diversely.

---

[2]This can be replaced with a probability model.

[3]The center vector $\boldsymbol{w}_{c,g_k}^{(l)}$ indicates a representative vector of the group $g_k$ at the $l$-th level and it is equivalent to a mean vector of $\boldsymbol{w}_{s,g_k,i}^{(l)}\ \forall i$: $\mu(\boldsymbol{w}_{s,g_k,i}^{(l)}) = \frac{1}{|g_k|} \sum_i^{|g_k|} \boldsymbol{w}_{s,g_k,i}^{(l)}$. The center vector for the group $g_k$ can be determined by a parameter vector at the $(l-1) - th$ level using an adjust factor ($\epsilon$, $|\epsilon| < 1$): $\boldsymbol{w}_{c,g_k}^{(l,l-1)} = \epsilon \cdot \boldsymbol{w}_{g_{k'},i}^{(l-1)}$, where $\boldsymbol{w}_{g_{k'},i}^{(l-1)} \in \mathbb{S}^d_{\boldsymbol{w}_{c,g_k'}^{(l-1)}}$.

Using parameter vectors $\boldsymbol{w}_i$ and $\boldsymbol{w}_j$ in $\mathbb{R}^{d+1}$, we define a discrete distance metric using a sign function as follows:

$$D_h := \frac{1}{d} \sum_k^d sign(\boldsymbol{w}_i(k)) \cdot sign(\boldsymbol{w}_j(k)), \tag{8}$$

where $sign(x) := \begin{cases} 1, & \text{if } x \geq 0 \\ -1, & \text{otherwise} \end{cases}$, $-1 \leq D_h \leq 1$, and $\boldsymbol{w} = \{\boldsymbol{w}(k) \mid \forall k = 1, \ldots, d+1\} \in \mathbb{R}^{d+1}$. This is a normalized version of Hamming distance. For a ternary discrete, $sign(x) \in \{-1, 0, 1\}$ is used. In order to consider the discrete distance as an angular distance within $[0, 1]$, normalized one is defined as $D_{h01} := \frac{-D_h + 1}{2}$, where $0 \leq D_{h01} \leq 1$. The angular distance based on the above product can be rephrased as $\theta_{D_h} = D_{h01}{}^4$ where $0 \leq \theta_{D_h} \leq 1$.

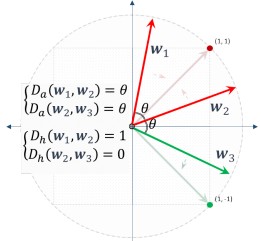

Figure 3: While the pairwise angular distances $D_a$ between a pair of vectors $\{\boldsymbol{w}_1, \boldsymbol{w}_2\}$ and $\{\boldsymbol{w}_2, \boldsymbol{w}_3\}$ are the same, the pairwise discrete product distances $D_h$ between vectors are different. To diversify a parameter space, the space with sign could be effective to recognize their difference.

As the discrete distance could underestimate to approximate the model distribution, we merge the above discrete distance metric with continuous angular distance metric ($\theta = \frac{1}{\pi} \arccos\left(\frac{\boldsymbol{w}_i \cdot \boldsymbol{w}_j}{\|\boldsymbol{w}_i\|\|\boldsymbol{w}_j\|}\right), 0 \leq \theta \leq 1$) into a single metric. We simply use the definition of Pythagorean means which consist of the arithmetic mean (AM), the geometric mean (GM), and the harmonic mean (HM). Pythagorean means using a pair of angular distances are defined as follows:

$$D_{\text{AM}} := \frac{\theta_{D_h} + \theta}{2}, \quad D_{\text{GM}} := \theta_{D_h}\theta, \quad D_{\text{HM}} := \frac{4\theta_{D_h}\theta}{\theta_{D_h} + \theta}. \tag{9}$$

In the angular distance[5] using a pair of angles $\{\theta_{D_h}, \theta\}$, a reversed form $1 - D_{\{\theta_{D_h}, \theta\}}$ is adopted to maximize an angle in the optimization formulation by minimization instead of $(\cdot)^{-s}$ where $s = 1, 2, \ldots$ which is used in Thomson problem that utilizes $s$-energy (Brauchart & Grabner, 2015). The cosine similarity using the pair of angles can be defined as follows:

$$D_{\cos(\text{AM})} := \cos\left(\frac{\theta_{D_h} + \theta}{2}\pi\right), D_{\cos(\text{GM})} := \cos\left(\theta_{D_h}\theta\pi\right), D_{\cos(\text{HM})} := \cos\left(\frac{4\theta_{D_h}\theta}{\theta_{D_h} + \theta}\pi\right), \tag{10}$$

then the cosine similarity functions are normalized with $\frac{\cos(\cdot) + 1}{2}$ to have a value within a range $[0, 1]$. Finally, Pythagorean means of cosine similarities can be calculated as follows:

$$D_{\text{AM}_{\cos}} := \frac{\cos\theta_{D_h}\pi + \cos\theta\pi + 2}{4}, D_{\text{GM}_{\cos}} := \frac{(\cos\theta_{D_h}\pi + 1)(\cos\theta\pi + 1)}{4}, D_{\text{HM}_{\cos}} := \frac{(\cos\theta_{D_h}\pi + 1)(\cos\theta\pi + 1)}{\cos\theta_{D_h} + \cos\theta + 2}. \tag{11}$$

Metric functions defined in (9), (10), and (11) satisfy metric conditions: non-negativity, symmetry, and triangle inequality. The distance using the above metric function between any two parameter points is bounded, because the hypersphere is a compact manifold.

### 3.3  GRADIENTS AND BACKPROPAGATION

As the sign function is not differentiable at the value 0, we adopt alternative backpropagation function. We adopt straight-through estimator (STE) (Bengio et al., 2013) in the backward path of the neural networks for the sign function in the discrete metric. The derivative of the sign function is substituted with $1_{|w| \leq 1}$ in the backward pass, known as the saturated STE. As the derivative of $\arccos(x)$ $\left(\frac{-1}{\sqrt{1-x^2}}\right)$ is undefined at the value $x = \pm 1$, we apply clamping to the cosine function to have $x \in [-0.99, 0.99]^6$ where $x = \cos(\theta\pi)$, and $0 \leq \theta \leq 1$.

## 4  EXPERIMENTS

### 4.1  EXPERIMENTAL SETUP

**Datasets**  We conduct experiments using four publicly available datasets including small size images (CIFAR-10 and CIFAR-100) and large size images (CUB200-2011 (Wah et al., 2011) and

---

[4]On the other hand, the angle can be considered as a cosine similarity directly, $D_h := \cos\theta_{D_h}\pi$. So as to get the angular distance, it needs an arccosine function $\theta_{D_h} = \frac{1}{\pi}\arccos D_h$. In summary, for the angle distance $\theta_{D_h}$, either "$D_{h01}$" or "$D'_{h01} = \frac{1}{\pi}\arccos D_h$" where $0 \leq D_{h01} \leq 1$, can be adopted.

[5]In $0 \leq \theta \leq 1$, the angle and its cosine value show an inverse relationship: $0 \leq \theta \leq 1 \to 1 \geq \cos\theta\pi \geq -1$.

[6]$x = \{0.99 \cdot 1_{x > 0.99}, x, -0.99 \cdot 1_{x < -0.99}\}$

Stanford-Cars (Krause et al., 2013b), shortly CUB200 and Cars hereafter respectively). CIFAR-10 dataset is used to validate effectiveness of the proposed metric without hierarchical information. Except CIFAR-10, we use two-level hierarchy pairs $\{parent, child\}$ for the proposed hierarchical regularization on three datasets. In Table 7 at Appendix, statistics of datasets in detail is provided.

**Deep neural network models and training setting**  We used the deep residual neural network (*resnet*) (He et al., 2016) with different sizes of parameters (weights) dependent on an image size of datasets. For a small size input ($32 \times 32$ pixels) of CIFAR-10 and CIFAR-100, we used *resnet* with smaller number of parameters (light models, resnet-20 (0.29M) and resnet-110 (1.73 M)) so as not to have redundant parameters leading to over-fitting. For a fine-grained input ($224 \times 224$ pixels) of CUB200 and Cars, the original resnet with larger number of parameters (heavy models[7], Resnet-18 (11.28M) and Resnet-50 (23.91M)) is used.

In training of the deep neural network with the hierarchical regularization, we assume that the global hierarchical structure is not given. Instead, stochastic or partial hierarchical structure is shown within each mini-batch by hierarchy pairs. Mini-batches, 512 for light models and 256 for heavy models, are used in the SGD optimizer. We applied the hierarchical regularization in the FC layer of the resnet. Even though SGD is known as an unbiased estimation, a stochastic hierarchical structure could affect the overall approximation performance upon the class distribution within the mini-batch. Settings in more detail are provided in Appendix.

**Baseline setting**  In terms of the regularization, we used three reference settings, denoted 'baseline', 'baseline+$l^2$', and 'E' for a reference. First, in 'baseline', we examined neural networks without regularization. Then, in 'baseline+$l^2$', we added $l^2$-norm minimization based regularization (which is equivalent to a weight decay in SGD setting (Zhang et al., 2019)) via $\lambda_f \sum_k \|\boldsymbol{w}_k\|$, where $\boldsymbol{w}_k \in \boldsymbol{W}$ and $\lambda_f > 0$ to 'baseline'. Lastly, in 'E', energy minimization (Liu et al., 2018), diversity promoting regularization without hierarchical information, via $\lambda_e \sum_{i \neq j} d(\boldsymbol{w}_i, \boldsymbol{w}_j)$, where $d(\cdot, \cdot)$ is a pairwise distance and $\lambda_e > 0$, is added to baseline+$l^2$. When the proposed hierarchical regularization (Eq. 5) is applied to the FC layer, we denoted it as 'H'. Hence, the regularization is incrementally adopted to the baseline: $l^2$+'E'+'H'. In terms of a distance metric, we compared the proposed metric with other continuous metrics such as Euclidean distance with the unit-length projection ('N-euclidean2', $\sum_{i \neq j} \| \frac{\boldsymbol{w}_i}{\|\boldsymbol{w}_i\|} - \frac{\boldsymbol{w}_i}{\|\boldsymbol{w}_i\|} \|^{-2})$[8], angular distance ('angular2', $\sum_{i \neq j} \arccos \left( \frac{\boldsymbol{w}_i \cdot \boldsymbol{w}_j}{\|\boldsymbol{w}_i\| \|\boldsymbol{w}_j\|} \right)^{-2}$) where '2' is from Riesz $s$-energy shown higher accuracy, and cosine similarity ('cosine', $\sum_{i \neq j} \frac{\boldsymbol{w}_i \cdot \boldsymbol{w}_j}{\|\boldsymbol{w}_i\| \|\boldsymbol{w}_j\|}$).

## 4.2 RESULTS

**Low resolution object classification**  In Table 1, we showed test accuracy (%) using different metrics with energy minimization regularization without the hierarchical regularization[9] along a weight decay ($l^2$-norm) using CIFAR-10 and resnet-20. The 'baseline' method is examined without ($\times$ in Table 1) or with ($\circ$ in Table 1) the weight decay ($l^2$). The discrete angular metric based regularization ($D_h^{ter}$ (ternary code), $D_h^{bin}$ (binary code), $D_{\cos \text{(HM)}}$, and $D'_{\text{HM}_{\cos}}$) on the baseline can improve the generalization performance in term of test accuracy compared to the other continuous metrics such as N-euclidean2, angular2, and cosine. Due to the unit-length projection in N-euclidean2, their performance is comparable to that other angular metrics. $D'$ denotes an arccosine adopted discrete distance used as $D'_h$ (see footnote 4 in Section 3.2). Regularization methods are applied over all layers except Batch Normalization (BN) layers. As explained in (van Laarhoven, 2017; Zhang et al., 2019), $l^2$-norm minimization based regularization (weight decay) showed much improvement due to their effective learning rate in SGD.

As shown in Table 2, the regularization with proposed metrics shows better accuracy performance than that of the baseline on both resnet-20 and resnet-110 using CIFAR-100 dataset. Comparing to the baseline+$l^2$, with energy minimization 'E' regularization, metrics such as $D_h^{ter}$, $D_{\cos(\text{AM})}$, and $D'_{\cos(\text{HM})}$ showed better performance than other metrics. If the proposed hierarchical 'H' regular-

---

[7]which is available at the pytorch library

[8]$= \sum_{i \neq j} \left( 2 - 2 \frac{\boldsymbol{w}_i \cdot \boldsymbol{w}_j}{\|\boldsymbol{w}_i\| \|\boldsymbol{w}_j\|} \right)^{-1}$

[9]This is because that hierarchy information is not included in CIFAR-10.

Table 1: Test accuracy (%) using different metrics on energy minimization without hierarchy information along $l^2$-norm minimization (weight decay) on CIFAR-10 and resnet-20.

| metric \ $l^2$ | × | ∘ |
|---|---|---|
| baseline | 90.34 | 92.21 |
| N-euclidean2 | 90.93 | 92.35 |
| angular2 | 90.47 | 92.38 |
| cosine | 90.53 | 92.40 |
| $D_h^{ter}$ | 90.67 | 92.48 |
| $D_h^{bin}$ | 90.67 | 92.48 |
| $D_{\cos(HM)}$ | 90.84 | 92.93 |
| $D'_{HM_{\cos}}$ | 90.94 | 92.69 |

Table 2: Test accuracy (%) using different metrics on energy minimization ('E') and hierarchical regularization ('H') on CIFAR-100: 'E' / 'E+H' respectively.

| metric \ db | resnet-20 | resnet-110 |
|---|---|---|
| baseline | 63.86 | 62.02 |
| baseline+$l^2$ | 68.03 | 72.90 |
| N-euclidean2 | 67.59 / 68.65 | 73.95 / 73.96 |
| angular2 | 67.83 / 67.76 | 74.40 / 73.89 |
| cosine | 68.11 / 68.45 | 73.37 / 73.37 |
| $D_h^{ter}$ | 68.44 / 68.68 | 73.73 / 73.97 |
| $D_h^{bin}$ | 68.52 / 68.69 | 73.97 / 74.26 |
| $D_{AM}$ | 68.58 / 68.86 | 73.43 / 73.50 |
| $D_{\cos(AM)}$ | 68.58 / 68.60 | 73.14 / 73.65 |
| $D'_{\cos(AM)}$ | 67.57 / 68.36 | 73.14 / 73.72 |
| $D'_{\cos(HM)}$ | 68.62 / 68.65 | 73.07 / 73.65 |

Table 3: Test accuracy (%) using different metrics on energy minimization ('E') and hierarchical regularization ('H') on CUB200: 'E' / 'E+H' respectively.

| metric \ db | Resnet-18 | Resnet-50 |
|---|---|---|
| baseline | 72.17 | 74.21 |
| baseline+$l^2$ | 72.29 | 74.05 |
| N-euclidean2 | 72.61 / 75.99 | 73.49 / 76.14 |
| angular2 | 72.43 / 76.11 | 73.55 / 76.66 |
| cosine | 72.12 / 75.64 | 73.26 / 76.85 |
| $D_h^{ter}$ | 72.58 / 75.99 | 73.57 / 76.37 |
| $D_h^{bin}$ | 72.55 / 76.21 | 73.57 / 76.99 |
| $D_{AM}$ | 73.04 / 76.02 | 73.88 / 75.95 |
| $D_{AM_{\cos}}$ | 72.31 / 76.14 | 73.59 / 77.32 |
| $D'_{AM_{\cos}}$ | 72.28 / 75.37 | 72.42 / 74.12 |
| $D_{GM}$ | 72.90 / 76.35 | 74.16 / 75.30 |
| $D'_{HM_{\cos}}$ | 72.55 / 76.11 | 74.64 / 76.94 |
| $D'_{\cos(HM)}$ | 72.55 / 76.32 | 72.86 / 76.56 |

Table 4: Test accuracy (%) using different metrics with energy minimization ('E') and hierarchical regularization ('H') on Cars: 'E' / 'E+H' respectively.

| metric \ db | Resnet-18 | Resnet-50 |
|---|---|---|
| baseline | 85.10 | 87.99 |
| baseline+$l^2$ | 85.58 | 87.92 |
| N-euclidean2 | 85.48 / 85.56 | 87.96 / 87.97 |
| angular2 | 85.11 / 85.13 | 88.34 / 87.85 |
| cosine | 85.57 / 85.73 | 88.01 / 87.86 |
| $D_h^{ter}$ | 85.35 / 85.99 | 85.35 / 88.07 |
| $D_h^{bin}$ | 86.22 / 85.99 | 88.32 / 88.14 |
| $D_{AM}$ | 85.66 / 85.66 | 88.39 / 88.11 |
| $D_{AM_{\cos}}$ | 85.52 / 86.05 | 87.92 / 88.57 |
| $D'_{AM_{\cos}}$ | 85.76 / 86.43 | 88.07 / 87.96 |
| $D'_{\cos(AM)}$ | 85.54 / 85.52 | 88.22 / 88.13 |

ization is applied, the generalization performance is further improved in most metric cases on both resnet-20 and resnet-110. As the binary metric shows a better performance than that ternary, we adopt binary discretization for proposed angular metrics ($D_{\bullet}$, $D_{\cos(\bullet)}$, and $D_{\bullet_{\cos}}$) in experiments.

**Fine-grained visual categorization**   In this experiment, we used two large size image datasets with a fine-grained category. One is with birds (CUB200) and another is with cars (Cars) that focus on single species of objects. A hierarchy of CUB200 is defined by the academic expert on birds whereas that of Cars is categorized manually based on a name of cars by a non-expert. Moreover, the rate between the number of superclass (parent) per subclass of CUB200 (0.35) is much larger than that of Cars (0.0459) (as shown in Table 7 at Appendix). That rate of Cars is even smaller than that of CIFAR-100.

As shown in Table 3, the proposed hierarchical regularization significantly improved the test accuracy over all metrics on both Resnet-18 and Resnet-50. Compared to CUB200, as shown in Table 4, the improvement of the proposed method is not that significant for Cars dataset. This is because CUB200 dataset has not only better hierarchical categorization by the expert but also more diversified pairs between the superclass and the subclass.

**Ablation study**   In Table 5, we showed how metrics affect the generalization performance using resnet-20 and CIFAR-100 dataset. The proposed method with different proposed metrics showed significantly improved performance compared to the baseline+$l^2$. Individual means (AM, GM, and HM) show different improvement patterns.

In Table 6, we showed how heterogeneous metrics on convolutional (conv.) layers (with the energy minimization, 'E') and fully connected (FC) layers (with the hierarchical 'H' regularization, 'H') affect the generalization performance using resnet-20 and CIFAR-100 dataset. The cases applying hierarchical regularization showed better performance than that the baseline applying only energy minimization regularization. In this experiment, a combination GM and HM shows a better improvement than that other combinations.

Table 5: Test accuracy (%) using different metrics on energy minimization ('E') and hierarchical regularization ('H') on CIFAR-100 and resnet-20.

| metric | |
| --- | --- |
| baseline+$l^2$ | 68.03 |
| $D'_{AM}$ | 68.64 |
| $D'_{GM}$ | 68.70 |
| $D'_{HM}$ | 68.80 |
| $D_{AM_{cos}}$ | 69.24 |
| $D_{GM_{cos}}$ | 68.55 |
| $D_{HM_{cos}}$ | 68.77 |
| $D'_{AM_{cos}}$ | 68.96 |
| $D'_{GM_{cos}}$ | 69.00 |
| $D'_{GM_{cos}}$ | 68.83 |

Table 6: Test accuracy (%) using heterogeneous metrics at Conv. layers and FC layer respectively, on energy minimization ('E') and hierarchical regularization ('H') on CIFAR-100 and resnet-20.

| metrics (in conv., in FC) | |
| --- | --- |
| baseline+$l^2$ | 68.03 |
| $(D_{GM}, null)$ | 68.22 |
| $(D_{GM}, D_{AM})$ | 68.58 |
| $(D_{GM}, D_{GM})$ | 68.62 |
| $(D_{GM}, D_{HM})$ | 69.04 |
| $(D_{GM}, D_{AM})$ | 68.62 |
| $(D_{GM}, D_{GM})$ | 68.65 |
| $(D_{GM}, D_{HM})$ | 68.70 |

## 5 RELATED WORKS

Promoting of diversity in an embedding space or model parameters is an widely adopted strategy in machine learning related area to improve the generalization performance (Cogswell et al., 2016; Yang et al., 2019; Li et al., 2012; Ratzlaff & Fuxin, 2019; Xie et al., 2017b; 2018; 2017a; Liu et al., 2018). This diversity based strategy can be applied in a variety of levels such as in a feature level (Cogswell et al., 2016; Xie et al., 2018), in a projection parameter level (Xie et al., 2017a; Liu et al., 2018), in a model ensemble level (Zhou et al., 2018; Ratzlaff & Fuxin, 2019), in a latent space model level (Ratzlaff & Fuxin, 2019; Liu et al., 2018), and in a generative model level (Yang et al., 2019; Ratzlaff & Fuxin, 2019). Among these approaches, some regularization methods require additional optimization effort. Enlarging pairwise distance between features (Xie et al., 2018) requires computational efforts due to their covariance matrix. In (Xie et al., 2017b), unit-eigenvalue is utilized via singular value decomposition. To optimize a direction and magnitude of the parameter vector alternatively, an alternating direction method of multipliers (ADMM) (Xie et al., 2017a) is used.

In terms of learning on a hypersphere, (Liu et al., 2017) proposed that hyperspherical convolution (SphereCov) replaces the traditional inner-product based convolution in order to conduct learning angular representation. On a hypersphere, regularization via promoting diversity of parameters has been proposed (Liu et al., 2018) based on Minimum Hyperspherical Energy.

In terms of hierarchical learning, (Nickel & Kiela, 2017; Ganea et al., 2018) proposed to apply hyperbolic space to embed the data in deep neural networks. They showed that learning representations in the hyperbolic space is effective to the data preserving a latent hierarchy compared to in the Euclidean space. Different from these works which are aimed for representation learning, our proposed method is focused on regularization learning using explicit (predefined) hierarchy information.

## 6 CONCLUSION

We proposed the regularization method, which maximizes a pairwise distance between parameters preserving a hierarchical structure. To define a hierarchical parameter space, we reformulated the topology space with multiple hyperspheres. In each hypersphere, a projection is parameterized by the surface parameter with the center parameter. By imposing a maximum distance between hierarchical parameters, diversity of parameters preserving semantic structure was promoted. For the distance metric in multiple separated spaces, we proposed a discrete metric integrated with a continuous metric. Extensive experiments using publicly available datasets (CIFAR-10, CIFAR-100, CUB200-2011, and Stanford Cars), the deep neural network with our proposed regularization method showed superior classification performance, especially when the number of super-classes is large. For further exploration in future, our proposed method can be combined with hierarchical representation learning such as hyperbolic (or Poincaré) embeddings (Nickel & Kiela, 2017; Ganea et al., 2018).

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

## APPENDIX

**Dataset acquisition details** CIFAR-100 dataset provides a pair of superclass-subclass labels. In CUB200 dataset, the pairs can be extracted from their filename. In Cars dataset, we parsed each subclass label to one of nine superclass vehicle types, such as "Sedan", "SUV", "Van" and etc., following (Krause et al., 2013a).

Table 7: Statistics of benchmark datasets

| Dataset | #classes $\{pa, ch\}$ | #train | #test | input size | #samples /class | #super /subclass |
|---|---|---|---|---|---|---|
| CIFAR-10 | $\{1, 10\}$ | 50,000 | 10,000 | $32 \times 32$ | 5000.00 | 0.1000 |
| CIFAR-100 | $\{20, 100\}$ | 50,000 | 10,000 | $32 \times 32$ | 500.00 | 0.2000 |
| CUB200 | $\{70, 200\}$ | 5,994 | 5,794 | $224 \times 224$ | 29.97 | 0.3500 |
| Cars | $\{9, 196\}$ | 8,144 | 8,041 | $224 \times 224$ | 41.55 | 0.0459 |

**Deep neural network models and training details** First, resnet-20 (0.29M) and resnet-110 (1.73 M), which include a combination of Basic blocks with output channels $[16, 30, 64]$ are adopted for light models. An input dimensionality of the fully connected (FC) layer (a classier) is 64 for both resnet-20 and resnet-110. Second, heavy models are adopted such as Resnet-18 (11.28M[10]) and

---

[10]Dependent on the number of classes and the corresponding center parameters, the size could variate (e.g. 11.42M for CUB200, 11.31M for Cars).

Resnet-50 (23.91M) which consists of the basic blocks (Resnet-18) or the bottleneck blocks with output channels [64, 128, 256, 512] in Conv. layers. An input dimensionality of the FC layer is 512 for Resnet-18 and 2048 for Resnet-50.

Networks are optimized with SGD for both light and heavy models: we fixed 1) the weight initialization with Random-Seed number '0' in pytorch, 2) learning rate schedule [0.1, 0.01, 0.001], 3) with momentum 0.9, 4) regularization: $l^2$-norm minimization with $\lambda_f = 0.0005$, 5) energy minimization $\lambda_e = \{0.1, 1\}$, and hierarchical minimization $\lambda_l = 0.1 \times \{1, 5\}$. All the regularization is not applied to the parameters of Batch Normalization (BN) layers, since BN is also a regularization function. A bias term in the FC layer is not used. The images in training and test, images are resized to 256 size. The images is cropped with $224 \times 224$ size at random location in training and at center location in test. Horizontal flipping is applied in training. The light models are trained from scratch without the pretrained weights for 300 epochs. The heavy model is trained using pretrained model provided by pytorch library[11] with 100 epochs. The experiments are conduced using GPU "NVIDIA TESLA P40".

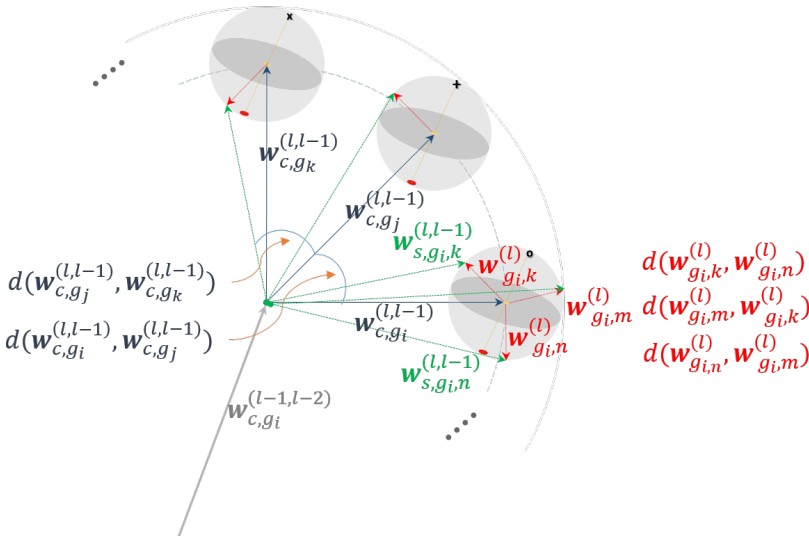

Figure 4: $\mathcal{R}_{l,p}(\boldsymbol{W}_{s,g_i}^{(l,l-1)}, \boldsymbol{w}_{c,g_i}^{(l,l-1)}) = d(\boldsymbol{w}_{g_i,n}^{(l,l-1)}, \boldsymbol{w}_{g_i,m}^{(l,l-1)}) + d(\boldsymbol{w}_{g_i,m}^{(l,l-1)}, \boldsymbol{w}_{g_i,k}^{(l,l-1)}) + d(\boldsymbol{w}_{g_i,k}^{(l,l-1)}, \boldsymbol{w}_{g_i,n}^{(l,l-1)}) + \dots$ corresponds to Eq (6) and $\mathcal{R}_{l,c}(\boldsymbol{w}_{c,g_k}^{(l,l-1)}) = d(\boldsymbol{w}_{c,g_i}^{(l,l-1)}, \boldsymbol{w}_{c,g_j}^{(l,l-1)}) + d(\boldsymbol{w}_{c,g_j}^{(l,l-1)}, \boldsymbol{w}_{c,g_k}^{(l,l-1)}) \dots$ corresponds to Eq (7)

***Constraints* in Eq. 4**    $\mathcal{C}_l := \sum_k \lambda_k \mathcal{C}_{l,k}$ can be given as follows:

1. *Constraint 1* ($\mathcal{C}_1$): This constraint describes that a radius of an inner sphere must be smaller than that of its outer sphere.
$r^{(l-1)} - r^{(l)} \geq 0 \Rightarrow \|\boldsymbol{w}^{(l-1)} - \boldsymbol{w}^{(l)}\| = \|\boldsymbol{w}_s^{(l-1)} - \boldsymbol{w}_c^{(l-1)}\| - \|\boldsymbol{w}_s^{(l)} - \boldsymbol{w}_c^{(l)}\| \geq 0.$

2. *Constraint 2* ($\mathcal{C}_2$): This constraint describes that a center of an inner sphere must be located within its outer sphere.
$r^{(l-1)} - (\|\boldsymbol{w}_c^{(l,l-1)}\| + r^{(l)}) \geq 0 \Rightarrow r^{(l-1)} - (\|\boldsymbol{w}_c^{(l-1)} - \boldsymbol{w}_c^{(l)}\| + r^{(l)}) = \|\boldsymbol{w}_s^{(l-1)} - \boldsymbol{w}_c^{(l-1)}\| - (\|\boldsymbol{w}_c^{(l-1)} - \boldsymbol{w}_c^{(l)}\| + \|\boldsymbol{w}_s^{(l)} - \boldsymbol{w}_c^{(l)}\|) \geq 0.$

3. *Constraint 3* ($\mathcal{C}_3$): This constraint describes that a margin between spheres must be larger than zero.
$\|\boldsymbol{w}_c^{(l,l-1)}\|(2 - 2\cos\theta)^{0.5} - 2r^{(l)} \geq 0 \Rightarrow \|\boldsymbol{w}_c^{(l)}\|(2 - 2\frac{\sum_{i \neq j} \boldsymbol{w}_c^{(l),i} \cdot \boldsymbol{w}_c^{(l),j}}{\|\boldsymbol{w}_c^{(l)}\|^2})^{0.5} -$

[11] from https://download.pytorch.org/models/resnet18-5c106cde.pth, https://download.pytorch.org/models/resnet50-19c8e357.pth respectively

$2\|\boldsymbol{w}_s^{(l)} \quad - \quad \boldsymbol{w}_c^{(l)}\|$, where $\|\boldsymbol{w}_c^{(l,l-1)}\|(2 - 2\cos\theta)^{0.5} \quad = \|\boldsymbol{w}_c^{(l,l-1)}\|(r^{(l-1)}\sin\theta^2 - (r^{(l-1)} - r^{(l-1)}\cos\theta)^2)^{0.5}.$

