# OpenReview forum: "Deep Hierarchical-Hyperspherical Learning (DH^2L)"
_ICLR.cc/2020/Conference — Reject_

### Official Review · AnonReviewer1 · 2019-10-21
**Official Blind Review #1**

**Rating:** 3

**Review:**

The paper proposes a hierarchical regularization framework based on hierarchical hyperspheres. In particular, the paper tackles the problem of diversity promoting learning. Following (Liu et al., 2018), pairwise distances between parameters on hyperspheres are used in the regularization framework.
The topology of the parameter space is reformulated with multiple hypersphere spaces which are each defined by two parameters: the centroid of a sphere and its surface vector. Multiple strategies involving hierarchical hyperspherical structures are proposed in Section 3 (continuous and discrete).
The relevance of the proposed method is experimentally demonstrated on different computer vision datasets.


I vote for reject for the following reasons:
- The paper is hard to read in general. Although the method section is understandable, its readability could be improved because each method currently just looks like a succession of equations. The paper also does not really give an intuition of why (or what contexts) one of the proposed regularizers would be better than the others.
- The reported (test accuracy) scores do not seem significantly better than the l2 baseline: none of the reported scores beats the l2 baseline by at least 1 percent, and it is unclear how that difference is measurable. How many splits/different initalizations were used? Why not give standard deviation over different test splits? etc... Given the fact that the improvements do not seem significant compared to a single baseline, a proper evaluation with standard deviation should be provided.
- Although the paper cites (Liu et al., 2018) as motivation for their framework, why does the proposed method does not compare to the other related work (i.e. works by Xie)?

As a side note, the paper seems to motivate the use of multiple spherical spaces to represent hierarchies. Recent work in machine learning has shown the advantage of using hyperbolic geometry [A,B] to represent trees, hence hierarchies.


[A] Nickel and Kiela, Poincaré Embeddings for Learning Hierarchical Representations, NIPS 2017
[B] Ganea et al., Hyperbolic Neural Networks, NeurIPS 2018


======= after the rebuttal

I have read carefully the updated manuscript, other reviews and rebuttal.
My score does not change since the proposed method does not seem to improve much compared to a simple weight decay (baseline + l2 regularization). The motivation of using the method for a very small improvement is not convincing. The "well-defined hierarchical information which is categorized by an expert as mentioned in the manuscript" can also be exploited by hyperbolic representations and should then also be compared (as baseline).

**Experience Assessment:**

I have read many papers in this area.

**Review Assessment: Checking Correctness Of Derivations And Theory:**

I assessed the sensibility of the derivations and theory.

**Review Assessment: Checking Correctness Of Experiments:**

I carefully checked the experiments.

**Review Assessment: Thoroughness In Paper Reading:**

I read the paper thoroughly.

---

> ### Author Response · Authors · 2019-11-11
> **Response to the review from AnonReviewer1**
>
> Thanks for your valuable review and constructive feedback!
>
> Q1. Improve readability.
>   A: We have revised the manuscript rigorously. 1) settings of compared methods and tables in the experimental section, 2) some equations concisely in the method section, 3) made sentences concisely, 4) inserted more related references (reviewer mentioned) in the related work section, 4) concise conclusions, and so on.
>
> Q2. An intuition of why (or what contexts) one of the proposed regularizers would be better than the others.
>   A: We followed a motivation of energy minimization (Liu's work) or angular constraints (Xie's work) which maximizes a margin between parameters for promoting diversity is an effective way of regularization. Based on this, we applied joint discrete and continuous metrics for a hierarchical space which is a combination of discrete space (between groups) and continuous space (within an individual group). Empirically, we showed that the proposed metrics with hierarchical settings can be effective in a hyperspherical space. And if the hierarchy is well defined, it shows significantly improved generalization performance.
>
> Q3. Beats the l2 baseline by at least 1 percent.
>   A: Note that our proposed method showed performance improvement orderly on all datasets. The improvement could not be large since the regularization is incrementally applied to the baseline (l2+‘E’+‘H’). In addition, 'l2' is applied to all layers but H is applied to only FC layer. Aspects of improvement is different dependent on datasets. Especially, on CUB200 where the hierarchical information is well defined, our proposed method showed significant improvement (+3%).
>
> Q4. How many different splits/initializations (with standard deviation).
> A: We agree with the reviewer's point of view that results with standard deviation is better to provide. But, single standard split for training set and test is given by datasets, so multiple splits are not available from the datasets we used (statistics in detail is shown in Appendix). As mentioned in appendix, we used the initialization has been fixed with Radom-Seed number 0 in pytorch (by using "torch.manual_seed(0), torch.backends.cudnn.deterministic = True, torch.backends.cudnn.benchmark = False", all parameters were fixed available. e.g. weight initialization, training sample sequence). By the fixed random seed, within the limited computational source, it was one of the best options to conduct the experiment with multiple comparisons between methods/metrics. As we think repeated trials with multiple different initialization in SGD setting should be meaningful, it will be added in appendix.
>
> Q5. Comparison to other related work (i.e. works by Xie).
>   A: We appreciate the reviewer for pointing out this comparison matter. One (Xie et al., 2017a) of Xie et al.'s works ((Xie et al., 2017b), (Xie et al., 2018), (Xie et al., 2017a)) was adopted already for a comparison in the manuscript. A pairwise angular constraint in (Xie et al., 2017a) is equivalent to energy minimization regularization ('E') which is included in our comparison except applying of a maximum angle value (in the paper $\tau$) and ADMM optimization for that constraint with $\tau$. The parameter $\tau$ which controls the level of diversity is another hyperparameter to find the optimum by found via an empirical verification. So, it seems more effective to use the constraint without this parameter. We will also insert happily this explanation into our manuscript if the reviewer thinks it needs. The other works were not directly related to our method. First, the work in (Xie et al., 2018) is focused on learning on the embedding space which is using both an input and a projection parameter while our work is learning on the projection parameter. Second, the work in (Xie et al., 2017b) is not directly applied a way of end-to-end manner to deep neural networks because of the eigen-decomposition, where an additional procedure is required.
>
> Q6. The advantage of using hyperbolic geometry.
>   A: [A,B] are quite useful references for hierarchical learning research. Our method is focused on parameter regularization while hyperbolic function based networks [A, B] are focused on representation learning. As hierarchical representation learning via hyperbolic spaces in our method would be a very useful strategy, we will apply it in a future work. We revised related works and conclusion sections by adding these references and related explanation.

---

### Official Review · AnonReviewer3 · 2019-10-24
**Official Blind Review #3**

**Rating:** 6

**Review:**

This paper proposes a regularization strategy motivated with principles of hierarchical, hyperspherical and discrete metric learning. Through regularization of as designed in level-wise, group-wise with the hierarchy of network, in their experiments with classification dataset, better performance are achieved with various distance.

Pros:
1: I think the paper is well organized and motivated, the regularization of parameters in deep neural network is one of the center problem for effective learning.
2: The proposed strategy is effective with their experiments, various datasets and objective metrics are adopted to validate the regularization.  Combination ablation study is sufficient.

Cons:
1: The paper is also related with several popular normalization strategies such as weight normalization/standardization, group/batch normalization. It would be more convincing that some comparison could be performed against these strategies.
2: There would be better to show its performance using larger dataset such as ImageNet or COCO detection.


**Experience Assessment:**

I have read many papers in this area.

**Review Assessment: Checking Correctness Of Derivations And Theory:**

I assessed the sensibility of the derivations and theory.

**Review Assessment: Checking Correctness Of Experiments:**

I carefully checked the experiments.

**Review Assessment: Thoroughness In Paper Reading:**

I read the paper at least twice and used my best judgement in assessing the paper.

---

> ### Author Response · Authors · 2019-11-11
> **Response to the review from AnonReviewer3**
>
> Thank you for the supportive review.
>
> Q1. several popular normalization
>   A: We agree with the review's comment. Among the popular normalization strategies the reviewer mentioned, the batch normalization is applied to the baseline (resnet by default). We clarify this in the revised manuscript. As our proposed method is experimented based on the baseline, the effect of batch normalization on the proposed regularization is shown. Even for a comparison between batch normalization and l2 regularization, interpretation and empirical results are slightly different between reference papers [Hoffer et al. 2018], [Zhang et al. 2019], [Laarhoven 2017]. We think a comparison between existing normalization strategies and our proposed regularization could be explored rigorously as a single topic in a future work. We can insert this explanation into our revised manuscript if the reviewer thinks so.
>
>   [Hoffer et al. 2018] "Norm matters: efficient and accurate normalization schemes in deep networks". Elad Hoffer, Ron Banner, Itay Golan, Daniel Soudry (NIPS2018).
>   [Zhang et al. 2019]  "Three Mechanisms of Weight Decay Regularization", Guodong Zhang and Chaoqi Wang and Bowen Xu and Roger Grosse (ICLR 2019).
>   [Laarhoven 2017] "L2 Regularization versus Batch and Weight Normalization", Twan van Laarhoven, Arxiv preprint.
>
> Q2. ImageNet or COCO detection
>   A: We appreciate the reviewer for suggesting this. Parsing of hierarchical label from the datasets is another task required much time. We planned to use ImageNet, unfortunately, parsing was not ready before the submission. We are currently preparing to apply ImageNet in the experiment.

---

### Official Review · AnonReviewer2 · 2019-10-26
**Official Blind Review #2**

**Rating:** 3

**Review:**

## Summary
The paper tackles the problem of promoting diversity in the weights of deep neural networks. The problem is interesting and useful. The paper argues that hierarchical learning and hyper spherical learning are important in addressing this problem. The paper provides experiments on CIFAR-10 and CIFAR-100 where the improvements of using such regularization is visible but not sufficiently significant.

## Contribution of the paper
1. The paper proposed a regularization to training neural networks with discrete angular distance metric on the weights.
2. The paper shows improved performance on CIFAR-10 and CIFAR-100.

## Overall feedback
I found the paper is well motivated and the proposed approach to be interesting. But I found the experimental validation a bit confusing. The improvments of the proposed approach also seems quite marginal. The contribution of different regularization terms is not understood clearly as well. So I am leaning towards rejection.

## Detailed feedback and questions for rebuttal
1. The writing could be improved significantly. I had a hard time to find exactly what different regularization terms are, e.g., E, H, L2. The paper could be more clear by clearly stating these regularization equations.
2. Please capitalize "eq. (1)" to "Eq. 1".
3. It seems there are three regularization E, L2 and H. But different tables show different combinations. For example, Table 1 has E and E+l2 while Table 2 has E and E+H and Table 3 has only E+H. Can you provide full results on all datasets on E, E+l2, E+H? Without seeing the full results it is hard to draw any conclusions.
4. Please correct the text "resnet-100" to "resnet-110" assuming you are using resnet-110.
5. It seems E+H improves marginally over E. Can you elaborate the explanation about it?
6. Why E+l2 improves so much (+2%) on CIFAR10?

**Experience Assessment:**

I have read many papers in this area.

**Review Assessment: Checking Correctness Of Derivations And Theory:**

I assessed the sensibility of the derivations and theory.

**Review Assessment: Checking Correctness Of Experiments:**

I assessed the sensibility of the experiments.

**Review Assessment: Thoroughness In Paper Reading:**

I read the paper at least twice and used my best judgement in assessing the paper.

---

> ### Author Response · Authors · 2019-11-11
> **Response to the review from AnonReviewer2**
>
> Thanks for your valuable review and constructive feedback!
>
> 1. Response to the overall feedback
>  Datasets and performance improvement: Thanks for a supportive review to the proposed approach.
> - In the experiment section, we conducted visual classification using four datasets not only CIFAR-10 and CIFAR-100, but also CUB200 and Cars. The improvement by hierarchical regularization with different metrics showed a different trend along datasets. Over datasets, the performance improvement seems dependent on the quality of hierarchical information. Especially, the performance improvement is significant (more than 3%) on CUB200 dataset. CUB200 has a well-defined hierarchical information which is categorized by an expert as mentioned in the manuscript. And this dataset has more superclasses are defined compared to the other dataset.
>
> - We observed that weight decay (L2 norm of the weights in the gradient descent setting) is very powerful, validated in [Zhang et al 2019]. In addition to this l2, our proposed hierarchical regularization is applied only to the FC layer whereas l2 and E (energy minimization) are applied over all layers. This could be one of reason why huge improvement is not shown in some datasets.
>
> [Zhang et al. 2019] "Three Mechanisms of Weight Decay Regularization", Guodong Zhang and Chaoqi Wang and Bowen Xu and Roger Grosse (ICLR 2019)
>
> - We revised the corresponding experimental paragraphs rigorously. For example, we added "Baseline setting" paragraph for clarifying comparisons. Responses in detail are shown below.
>
> 2. Responses to the detailed feedback
> Q1. Writing improvement. e.g., clearly stating regularization equations E, H, L2.
>   A: We have revised the manuscript further to be readable and understandable by a reader (reviewer). We added "Baseline setting" to state regularization equations clearly (please see the third paragraph of Section 4.1.). We revised experimental result section (Section 4.2) including tables correspondingly.
>
> Q2 (&Q4). Capitalize eq. to Eq., correct the text "resnet-100" to "resnet-110", and other typos.
>   A: We corrected it accordingly.
>
> Q3. Full results on all datasets on E, E+l2, E+H.
>   A: We appreciate the reviewer for pointing out this confusion at the previous tables. In Table 2, 3, and 4, we had provided the full results using E, E+l2, and E+H. For a better understanding, we inserted "Baseline setting" paragraph in the experimental section (Section 4.1) revised. The regularization is incrementally applied to the baseline (l2+‘E’+‘H’). That is, 1) baseline indicates no regularization applied, 2) baseline+l2 indicates l2 applied to 1), 3) 'E' indicates E applied to 2), 4) 'E+H' indicated 'H' applied to 3). Hence, we had shown full result for all dataset except CIFAR-10 which has no hierarchical information. We used CIFAR-10 dataset for evaluating the regularization performance along different proposed metrics without hierarchical regularization (H).
>
> Q5. E+H improves marginally over E.
>   A: We note that our proposed metrics are used in the E regularization. Regarding performance improvement (marginally) by H regularization, we note three reasons as follows, first, E is applied to all convolutional layers. H is applied only to the FC layer. Second, when the number of superclass is large which means more complicated hierarchy, it shows more improvement, more than 3 percent, (CUB200 > CIFAR-100 > Cars). Third, when the hierarchy information is clearer, more improvement: (CUB200: hierarchical categorized by bird experts, Cars: found a hierarchy by an unclear definition (based on a name of cars). For example, among nine superclasses (SUV, Sedan, Coupe, Convertible, Hatchback, Pickup, Minivan, Van, and Wagon), Hatchback, Wagon, and Coupe are very ambiguous to categorize them visually.
>
> Q6. E+l2 improves so much (+2%) on CIFAR10.
>   A: It is mainly because of l2-norm regularization. To clarify it, we revised Table 1, by indicating "with or without 'l2'". As mentioned above, l2 regularization is a weight decay term which realizes an effective learning rate. The weight decay term seems quite effective on datasets on CIFAR-10 and CIFAR-100 which consist of small image size (32 x 32) compared to larger size dataset (CUB200 and Cars, 224 x 224). Furthermore, as CIFAR-10 has ten classes only, a regularization term seems quite effective on this condition.
>
> Note that we have revised an order of tables, Table 3 and 4 became Table 5 and 6 respectively, and vice versa.

---

### Author Response · Authors · 2019-11-11
**For All Reviewers**

We appreciate reviewers for their valuable reviews and constructive feedback. We have addressed all of individual review's comments. Here, we summarize responses to reviewers' comments. Firstly, we have revised the manuscript rigorously for a readable experimental section: we have clarified settings of compared methods, revised result tables, and rearranged an order of paragraphs. We have revised other sections too. Secondly, We respond to the comment that the proposed method shows marginal performance improvement. We note that our proposed method showed performance improvement systematically on all datasets. Over datasets, the amount of performance improvement seems dependent on the quality of hierarchical information. Especially, the performance improvement is significant (more than 3%) on CUB200 dataset. More improvement is shown if more well-defined hierarchical information is used. Another reason is that the regularization is incrementally applied to the baseline, i.e. baseline, baseline+l2 (weight decay) + ‘E’ with proposed metrics + ‘H’ with proposed hierarchical regularization.

---

### Decision · Program_Chairs · 2019-12-19

**Decision:**

Reject

**Comment:**

The paper proposes a hierarchical diversity promoting regularizer for neural networks. Experiments are shown with this regularizer applied to the last fully-connected layer of the network, in addition to L2 and energy regularizers on other layers. Reviewers found the paper well-motivated but had concerns on writing/readability of the paper and that it provides only marginal improvements over existing simple regularizers such as L2. I would encourage the authors to look for scenarios where the proposed regularizer can show clear improvements and resubmit to a future venue.